# ActDR: Action Difference Reasoning via Keypoint Guided Tree Search

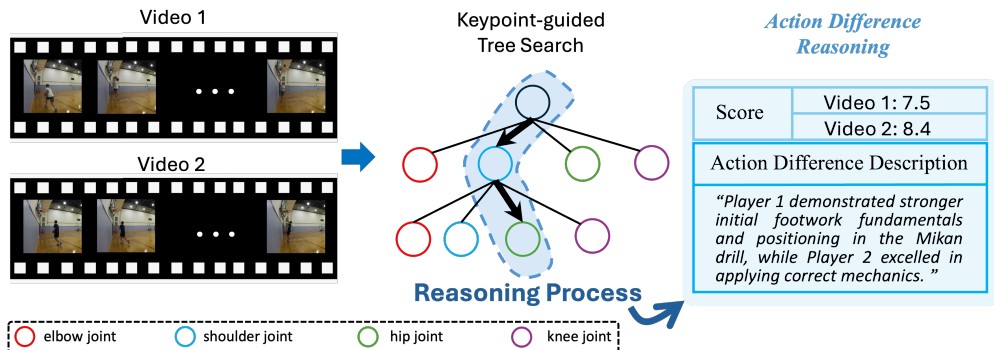

Figure 1: Overview of the action difference reasoning (ADR) task and our proposed Keypoint-guided Tree Search approach. Our model takes two videos doing the similar action as inputs, generating both *quantitative* score predictions and *qualitative* action difference descriptions between the two videos.

## Abstract

Analyzing fine-grained differences in skilled activities, such as sports or surgery, poses a significant challenge for computer vision, demanding both precise action understanding and domain-specific reasoning. While prior work has made progress in evaluating individual performance, existing methods fall short in comparing two similar actions (e.g., penalty kick in soccer) conducted by different performers and explaining *how* their actions differ. To address this gap, we introduce **Action Difference Reasoning (ADR)**, a novel task that jointly provides *quantitative* performance scores and *qualitative* explanations of inter-performer differences, enabling actionable feedback for improvement. To support this task, we construct the ADR dataset, built upon Ego-Exo4D dataset, comprising paired videos annotated with both performance scores and natural language descriptions of action differences. We further propose **KEPT**, a *keypoint guided tree search* framework that explicitly models the reasoning process behind performance differences by capturing fine-grained kinematic cues. Experiments on the ADR dataset show that KEPT significantly outperforms existing baselines, including large vision-language models, on both score prediction and action difference explanation. Moreover, our framework generalizes effectively to traditional Action Quality Assessment (AQA) settings, surpassing state-of-the-art approaches on benchmarks including JIGSAWS and FitnessAQA. Code, model and dataset will be released after the review process.

## 1 Introduction

Analyzing and comparing skilled activities (e.g., sports, surgery) is highly valuable for education and coaching, supporting trainees across different skill levels. For novice athletes, like beginner soccer players, precise analysis of actions (e.g., ball control, kicking) helps decompose complex movements into fundamental components. For advanced performers, fine-grained action analysis enables targeted feedback for skill refinement and performance optimization. Similar needs arise in surgical training, where junior surgeons benefit from detailed guidance on how to improve their technique. Crucially, effective training requires both quantitative assessment and qualitative

interpretation of fine-grained actions. However, analyzing actions in isolation is often insufficient. A more effective approach involves comparing the same actions across different performers to uncover subtle execution differences. These detailed comparisons reveal specific areas for improvement, providing actionable insights that enable performers to refine their techniques and accelerate skill development.

Prior work on analyzing skilled activities Xu et al. (2024b); Zhou et al. (2024); Majeedi et al. (2024); Bai et al. (2022); Yun et al. (2024); Xu et al. (2024a; 2022) has largely focused on evaluating individual performers in isolation. While effective for assessing performance quality, this approach lacks the ability to capture and explain fine-grained differences between performers' techniques. As a result, it limits the potential for comparative insights that are critical for identifying subtle execution variations—insights that can drive personalized feedback and targeted skill refinement. Recent work by Burgess et al. Burgess et al. (2025) introduced the video action differencing task, which compares how two individuals perform the same actions. However, their approach is limited to qualitative assessment only, without quantitative assessment.

To overcome the above limitations, we propose a new task: **Action Difference Reasoning (ADR)**. Unlike traditional score-based methods, ADR aims to provide both *quantitative* individual assessments and *qualitative* explanations of differences between performers' actions, offering deeper insight into skill variation. As shown in Figure 1, ADR takes as input two videos of different individuals performing similar actions and outputs a numerical score reflecting each performance quality along with language commentary describing key action differences. This dual-output formulation enables more comprehensive and interpretable analysis, supporting targeted feedback for skill development.

As no existing datasets support the task of Action Difference Reasoning (ADR), we introduce a new dataset, ADR, which is built on the Ego-Exo4D dataset Grauman et al. (2024). ADR focuses on exocentric videos from three sports domains: basketball, soccer, and rock climbing. For each sport, we select one exocentric video per player, annotated with expert-assigned performance scores and commentary. We then construct video pairs capturing the same sub-activity within each sport (e.g., mid-range shooting in basketball). For each pair, we use expert commentary to prompt a Large Language Model (LLM) to generate detailed descriptions of the action differences between performers, as illustrated in Figure 2. This structured dataset enables comprehensive ADR analysis by supporting both quantitative evaluation and qualitative reasoning about inter-performer differences.

The Action Difference Reasoning (ADR) task presents unique challenges: identifying fine-grained differences between performers' actions requires both precise motion understanding and domain-specific expertise. Accurate motion understanding involves capturing the spatiotemporal kinematic patterns of the performers. Meanwhile, domain expert, such as human referees, rely on structured visual reasoning processes guided by their specialized knowledge to identify subtle discrepancies. To effectively address ADR, it is essential to model these expert-like *visual reasoning processes* and incorporate *domain knowledge* into the system, enabling the model to distinguish nuanced action differences with both accuracy and interpretability.

Given that domain experts typically engage in structured visual reasoning by searching in their internal knowledge graph, we formulate visual reasoning as a *search process* and introduce the **KEPT** framework that combines keypoint-based motion analysis with tree search to model expert-like reasoning. In this framework, tree search serves as a sequential decision-making algorithm that explores a combinatorial space of possible reasoning trajectories, where each path represents a plausible sequence of shifts over spatiotemporal keypoint features on the human body. This process enables the model to discover suitable reasoning paths that align with expert judgments, allowing for both interpretable and fine-grained comparison of performers' actions. Our approach begins by applying tree search to explore sequential visual reasoning paths over keypoints, capturing fine-grained kinematic dynamics in a structured and interpretable manner. To enhance alignment with human expert judgment, we guide the learned reasoning paths using expert commentary, ensuring that the model captures semantically meaningful action differences. These reasoning path representations are then used to generate quantitative performance scores. In parallel, we compute a residual implicit keypoint graph embedding by contrasting the two performers' keypoint graphs, allowing the model to isolate subtle motion discrepancies. This fusion of structured visual reasoning and implicit relational cues enables the generation of qualitative natural language explanations that are both detailed and grounded in expert knowledge, resulting in a comprehensive ADR output. The major contributions of this paper are summarized as follows:

- We introduce a novel task, Action Difference Reasoning (ADR) for skilled activities, which not only generates score evaluations for individual performers but also provides detailed, expert-level commentary on action differences between players across multiple sports categories.

- We propose a new framework called KEPT, the Keypoint-guided Tree Search, designed to address the challenges of ADR tasks by visual reasoning processes essential to assessing action performance.

- To support ADR research, we construct a new dataset, *ADR*, which includes both score-based evaluations and expert-generated comments on action differences. Comprehensive experiments demonstrate that our approach significantly outperforms existing state-of-the-art methods on both this challenging ADR task and traditional AQA task, highlighting its effectiveness in detailed sports performance analysis.

## 2 ACTION DIFFERENCE REASONING DATASET

To support research on action difference reasoning, we constructed a new dataset, *ADR*, derived from the Ego-Exo4D dataset Grauman et al. (2024). *ADR* includes videos from three sports: soccer, basketball, and rock climbing. Soccer and basketball activities are divided into three sub-activities each, while rock climbing is represented by a single sub-activity. As detailed in Table 1, the basketball activities include Mid-Range Jump Shooting, Mikan Layup, and Reverse Layup, while soccer activities are categorized as Penalty Kick, Juggling, and Dribbling. To maintain clean scenarios, we selected one exocentric video per player from the original dataset, choosing only videos where no other unrelated individuals are present. The composition of each category and the number of videos per sport are shown in Table 1.

Figure 2: The process of data construction for action difference description generation with Gemini 2.5. We take the instruct prompt and expert comments on video 1 and video 2 as inputs to Gemini 2.5, generating the ground-truth action difference descriptions.

Since every player is commented by several experts with individual score evaluations, the ground-truth score for each player is obtained by averaging all the scores provided by experts. Expert comment for each video is also obtained by combining all expert comments. To construct the *ADR* dataset, we group videos in each sub-activity into video pairs, as the examples shown in Figure 2. Then the ground-truth action difference descriptions are generated by prompting Gemini 2.5 Gem (2025) with expert comments on each video. Figure 2 shows the process of constructing ground-truth action

| Category | Sub-activity | Num of Videos | | Num of Video Pairs | |
|---|---|---|---|---|---|
| | | train | test | train | test |
| Basketball | Mid-Range Shooting | 17 | 28 | 136 | 378 |
| | Mikan Layup | 26 | 23 | 325 | 253 |
| | Reverse Layup | 27 | 30 | 351 | 435 |
| Soccer | Juggling | 6 | 2 | 15 | 1 |
| | Penalty Kick | 21 | 3 | 210 | 2 |
| | Dribbling | 23 | 5 | 253 | 10 |
| Rock Climbing | Climbing | 70 | 155 | 2415 | 703 |

Table 1: Dataset Splits for each sports category. Both basketball and soccer contain three categories, while rock climbing contains only one category. The number of videos and video pairs for each sports category are listed.

difference descriptions. To ensure that the generated descriptions focus on action differences of players, the instruction prompt for Gemini 2.5 is designed to stress that the summary should include which part does which player do better, as shown in Figure 2. The number of video pairs generated for each sports category is also listed in Figure 1.

To validate the quality of action difference descriptions generated by Gemini 2.5, we conducted a systematic human evaluation using 50 randomly sampled video pairs from each sport category. A human evaluator with domain knowledge in each respective sport was tasked with determining which athlete demonstrated superior performance, based solely on the model-generated action difference descriptions. We then compared these human judgments against the ground-truth athlete scores to assess whether the generated descriptions accurately captured performance differentials. The evaluation revealed strong alignment between the descriptions and actual performance metrics, with 128 out of 150 video pairs ($85\%$) yielding judgments consistent with ground-truth score differentials. This substantial agreement rate confirms the high fidelity and reliability of the generated action difference descriptions in capturing detailed performance variations between athletes.

## 3 METHOD

Our KEPT approach is outlined in Figure 3, which consists of three major components: (1) Keypoint-guided Tree Construction; (2) Expert Knowledge Alignment; (3) Keypoint-guided Action Difference Reasoning.

### 3.1 TASK FORMULATION

We introduce the novel task called Action Difference Reasoning (**ADR**) that generates both quantitative score assessment and qualitative language descriptions on action differences between two performers doing the same action. Specifically, given two videos $V_1$ and $V_2$ that doing the same category of action $a$, ADR task requires to output both quantitative results and qualitative results:

- predicted scores $s_1$ and $s_2$ of the two corresponding videos (*Quantitative*).
- language descriptions $\mathcal{T}$ of the difference of the two videos (*Qualitative*).

Our objective is to learn a model $\mathcal{M}_\theta$ parametrized by $\theta$ that generates these two components: $s_1, s_2, \mathcal{T} = \mathcal{M}_\theta(V_1, V_2)$. During training, we have original videos, ground-truth scores and language descriptions of action difference. For testing, we only have videos, aiming to generate scores and language descriptions of action difference.

### 3.2 KEYPOINT GUIDED TREE SEARCH

Human referees employ specific sequential visual process strategies to accurately evaluate a performer's action, focusing on distinct regions of interest at different times. For instance, a referee might initially attend to upper body kinematics before shifting their gaze to lower body movements. The final assessment is then derived from this temporal sequence of visual fixations. Drawing an analogy to this sequential nature of human visual processing, we propose to model these visual process strategies as combinatorial search spaces. We formulate the visual processes of referees as such spaces and leverage keypoint-guided tree search to model and potentially predict their dynamic attention patterns.

As shown in Figure 3, our KEPT approach starts from keypoint-guided tree search to select an optimal reasoning path, followed by expert knowledge alignment and keypoint-guided action difference reasoning.

**Keypoint-Guided Tree Construction.** In the constructed search trees, each node represents a state (represented with circles of different colors in Figure 1), each edge represents a transition from one state to another. To start with, we first define the tree nodes based on the particular problem we are solving. In sports scenarios, we define tree nodes based on the four major joints that control human movement: *elbow*, *shoulder*, *hip*, and *knee*. Therefore, every tree node in sports scenarios consists of four possible actions. In surgical scenarios, since surgical actions are performed via surgical instruments, we define tree nodes based on the three major parts of instruments: *tip*, *body* and *tail*. Therefore, every tree node in surgical scenarios consists of three possible actions. We use sports scenarios as running examples and include illustrations on surgical scenarios in supplement materials.

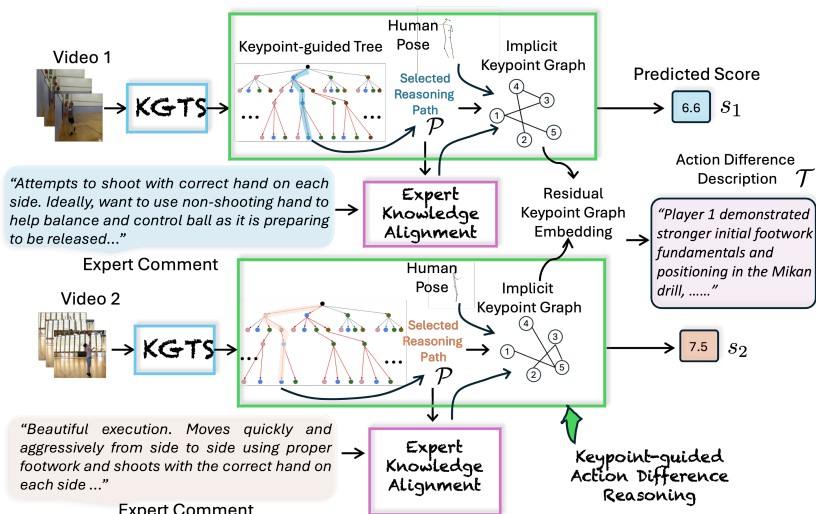

Figure 3: The pipeline of our KEPT framework, including Keypoint-guided tree search (KGTS), Expert Knowledge Alignment and Knowledge-guided Action Difference Reasoning.

We implement the keypoint-guided tree with a finite depth $K$. Each leaf node involves $K-1$ steps and corresponds to a unique trajectory through the assessment process, representing a sequence of consecutive. To represent each node, we utilize the three key physical information of each joint: *angle*, *speed*, and *position*. Formally, each node is represented in the form of $(a, v, p)$ where $a \in [0, 1]$ indicates angle, $v \in \mathbb{R}$ denotes the joint speed and $p = (x, y, z)$ indicates the position of the joint, where $x \in [0, 1]$, $y \in [0, 1]$ and $z \in [0, 1]$ are normalized 3D coordinates. This representation models the hierarchical relationships between major joints and their associated kinematic properties, providing a structured representation of the domain knowledge relevant to skilled human activities.

**Reasoning Path Selection.** As we have constructed the keypoint-guided tree, we learn the optimal reasoning path $l^*$ with:

$$l^* = \arg\max_{l \in T} Q(l), \tag{1}$$

where $l^*$ denotes the learned reasoning path with the highest score, $T$ is the number of leaf nodes in the keypoint-guided tree, which also corresponds to the number of reasoning paths, $Q(l)$ indicates the score value of each path.

To learn the score value $Q(l)$ of each reasoning path, we design a score network $\mathcal{R}_\psi$ (shown in the red block in Figure 4) based on the constructed keypoint-guided tree. Thus, $Q(l_i) = \mathcal{R}_\psi(l_i, V) \in [0, 1]$, where $i \in [1, T]$, $l_i$ indicates the $i$-th reasoning path, $V$ is the input video. For all the possible $K$ nodes ($K$ equals to the largest depth of keypoint-guided tree) in one reasoning path $l_i$, they are represented with their corresponding physical statistics of each joint, as illustrated in the above section. The score value $Q(l_i)$ for one path is obtained as:

$$\begin{aligned} Q(l_i) &= \text{Sigmoid}\big(\mathcal{F}_R(\text{MHCA}(f'_M, f'_V))\big) \in [0, 1], \\ f'_M &= \mathcal{F}_M(f_M) \in \mathbb{R}^{N \times d}, \\ f'_V &= \mathcal{F}_V(f_V) \in \mathbb{R}^{N \times d}, f_V = E_V(V) \in \mathbb{R}^{N \times d}, \end{aligned} \tag{2}$$

where MHCA indicates multi-head cross attention module in the Transformer block, $\mathcal{F}_R$, $\mathcal{F}_M$ and $\mathcal{F}_V$ are mapper functions implemented with Multi-Layer Perceptron (MLP), $E_V$ denotes an existing visual encoder, $d$ indicates the feature dimension, $f_M \in \mathbb{R}^{N \times h}$ represents the physical statistics (angle, speed and position) of keypoint-guided tree, $N$ is the number of video keyframes in a video input, $h$ is the size of physical statistics, $V$ indicates the video input. Therefore, we select the optimal reasoning path based on Eq 1.

Given that human referees tend to take similar visual reasoning processes while assessing the same actions, we introduce the KL divergence loss to measure the score value distributions between videos performing the same action:

$$\mathcal{L}_{\text{KL}} = \text{KL}(p_r^1, p_r^2), \tag{3}$$

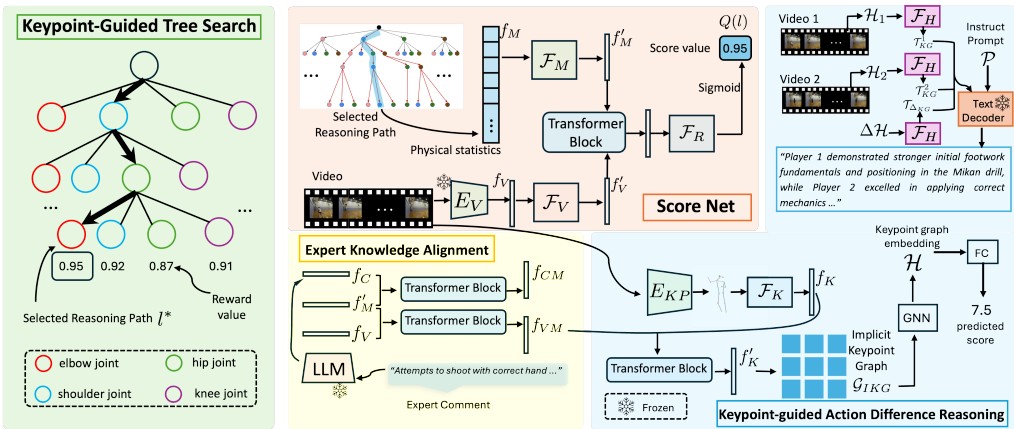

Figure 4: Detailed architecture of modules in the KEPT framework. There are three major components in KEPT: (1) keypoint-guided Tree Search that constructed with the learned score net; (2) Expert Knowledge Alignment; (3) Keypoint-guided Action Difference Reasoning. Each module is highlighted with different color.

where $p_r^1 \in \mathbb{R}^T$ and $p_r^2 \in \mathbb{R}^T$ indicate normalized score value distributions of the two videos with the same action categories, $p_r = \big(p(l_1), p(l_2), \cdots p(l_T)\big)$, $p(l_i) = \frac{Q(l_i)}{\sum_{i=1}^{T} Q(l_i)}$. KL divergence loss in Eq 3 enforces the score value distribution of two performers conducting the same actions to be close to each other.

### 3.3 Expert Knowledge Alignment

To learn better reasoning path representations $f'_M$, we further align $f'_M$ with expert commentary by incorporating expert knowledge. As shown in Figure 4, $f_{CM} = \text{MHCA}(f_C, f'_M) \in \mathbb{R}^{N \times d}$, $f_{VM} = \text{MHCA}(f_V, f'_M) \in \mathbb{R}^{N \times d}$, where $f_C \in \mathbb{R}^d$ and $f_V$ serve as query, $f_M$ serves as key and value in MHCA, $f_C$ denotes expert commentary representation extracted from a Large Language Model (LLM). Expert knowledge alignment is conducted by introducing the semantic loss function $\mathcal{L}_{\text{semantic}}$ to align visual representation $f_{VM}$ with language representation $f_{CM}$:

$$\mathcal{L}_{\text{semantic}} = 1 - \text{cos\_sim}(\bar{f}_{CM}, \bar{f}_{VM}), \tag{4}$$

where cos_sim indicates the cosine similarity between two embeddings; $\bar{f}_{CM} \in \mathbb{R}^d$, $\bar{f}_{VM} \in \mathbb{R}^d$ denote the average-pooled embeddings from $f_{CM} \in \mathbb{R}^{N \times d}$ and $f_{VM} \in \mathbb{R}^{N \times d}$.

### 3.4 Keypoint-guided Action Difference Reasoning

As we have learned the reasoning path representation $f'_M$ with expert knowledge alignment, we further update keypoint embedding $f_K$ into knowledge-enhanced keypoint embedding $f'_K$: $f'_K = \text{MHCA}(f_{VM}, f_K) \in \mathbb{R}^{P \times d}$, $f_K = \mathcal{F}_K(\mathbf{K}) \in \mathbb{R}^{P \times d}$, $\mathbf{K} = E_{KP}(V) \in \mathbb{R}^{P \times 3}$, where $\mathbf{K} \in \mathbb{R}^{P \times 3}$ indicates the extracted 3-D human keypoints from an existing keypoint extractor $E_{KP}$, $P$ denotes the number of keypoints.

**Action Score Prediction.** As shown in Figure 4, to capture the inter-dependencies among keypoints, we introduce an implicit knowledge graph (IKG) $\mathcal{G}_{\text{IKG}}$ based on knowledge-enhanced keypoint embedding $f'_K$. To generate the IKG, we start with an ordinary keypoint graph $\mathcal{G} \in \mathbb{R}^{P \times P}$, where each entry $g_{ij}$ ($i \in [1, P], j \in [1, P]$) is defined as follows: $g_{ij} = 1$ if keypoints $i$ and $j$ are connected by the human skeleton, and $g_{ij} = 0$ otherwise. To construct the implicit knowledge graph $\mathcal{G}_{\text{IKG}} \in \mathbb{R}^{P \times P}$, we invert the entries in $\mathcal{G}$, setting all positions with a value of 1 to 0. We then compute the probability for each remaining location in the graph as follows:

$$P(x_i, x_j | \theta) = \frac{1}{Z(\theta)} \exp\big(-\phi(x_i, x_j; \theta)\big),$$

$$Z(\theta) \triangleq \sum_{(x, x') \in \mathcal{P}} \exp\big(-\phi(x, x'; \theta)\big), \tag{5}$$

where $x_i \in \mathbb{R}^d$ and $x_j \in \mathbb{R}^d$ are individual keypoint embeddings obtained from the knowledge-enhanced keypoint embedding $f'_K$; $\mathcal{P}$ indicates all the poosible keypoint pairs; $Z(\theta)$ is the normalizing factor; $\phi(x, x'; \theta)$ denotes a mapper function (implemented with an MLP) parametrized with $\theta$ to learn the dependencies between keypoint embeddings $x$ and $x'$. To construct the $\mathcal{G}_{\text{IKG}}$, we set a threshold $\tau$ for the probability $P_{ij}$ obtained from Eq 5: $\{P_{ij} = 1 \text{ if } P_{ij} >= \tau; \ P_{ij} = 0 \text{ if } P_{ij} < \tau\}$. With both the original keypoint graph $\mathcal{G}$ and the implicit knowledge graph $\mathcal{G}_{\text{IKG}}$, we utilize graph neural network (GNN) to learn the final keypoint graph embedding $\mathcal{H} \in \mathbb{R}^{P \times d}$: $\mathcal{H} = \sigma\big(\mathcal{G}_{\text{IKG}}\big(\sigma(\mathcal{G}f'_K W)\big)W_{\text{IKG}}\big)$, where $\sigma$ indicates the nonlinear activation function; $W \in \mathbb{R}^{d \times d}$ and $W_{\text{IKG}} \in \mathbb{R}^{d \times d}$ are learnable weights in GNN. The predicted score $\hat{s}$ for each video segment is generated with a light-weight fully connected layer $\mathbf{FC}$: $\hat{s} = \frac{1}{K}\sum_{k=1}^{K} \mathbf{FC}(\mathcal{H})$, where $K$ denotes the number of segments for the entire video. The score prediction process is supervised with the score loss function $\mathcal{L}_{\text{score}}$:

$$\mathcal{L}_{\text{score}} = ||\hat{s} - s||_2^2, \tag{6}$$

where $s$ denotes the ground-truth score of the video clip. $\mathcal{L}_{\text{score}}$ aims to minimize the difference between the model-predicted action scores and the ground-truth values, ensuring that the model accurately evaluates action quality.

**Action Difference Description Generation.** For generating action difference descriptions, we regard it as comparing the differences of implicit keypoint graphs $\mathcal{G}_{IKG}$ between different videos. Firstly, keypoint graph embedding $\mathcal{H}$ of each video clip is applied with the keypoint graph token learning function $\mathcal{F}_H$ (implemented with an MLP) that maps $\mathcal{H}$ to keypoint graph token embeddings $\mathcal{T}_{KG}^1 = \mathcal{F}_H(\mathcal{H}_1) \in \mathbb{R}^{P \times d'}$, $\mathcal{T}_{KG}^2 = \mathcal{F}_H(\mathcal{H}_2) \in \mathbb{R}^{P \times d'}$, where $d'$ denotes the dimension of tokens. Secondly, we generate residual keypoint graph embedding $\Delta\mathcal{H} = [\mathcal{H}_1 - \mathcal{H}_2; \mathcal{H}_2 - \mathcal{H}_1]$, where $[;]$ denotes the concatenate operation. Thus, the token embedding $\mathcal{T}_{\Delta KG}$ of $\Delta\mathcal{H}$ is obtained by $\mathcal{T}_{\Delta KG} = \mathcal{F}_H(\Delta\mathcal{H}) \in \mathbb{R}^{P \times d'}$. We concatenate $\mathcal{T}_{KG}^1$, $\mathcal{T}_{KG}^2$, $\mathcal{T}_{\Delta KG}$ and the text embedding $f_{\mathcal{P}}$ of a short instruct prompt $\mathcal{P} \in \mathbb{R}^{d'}$, then apply a text decoder to generate the final descriptions of action differences $\mathcal{C}$:

$$\hat{\mathcal{C}} = \text{Text\_Decoder}([\mathcal{T}_{KG}^1; \mathcal{T}_{KG}^2; \mathcal{T}_{\Delta KG}; f_{\mathcal{P}}]), \tag{7}$$

where $\mathcal{P}$ is a short prompt to instruct LLM generating action difference descriptions (e.g., $\mathcal{P}$=`"Please generate the action difference between Player 1 and Player 2 given their respective videos."`). To monitor the text generation process, we extract features of both the ground-truth action difference description $\mathcal{C}$ and the generated action difference description $\hat{\mathcal{C}}$ with a LLM: $f_{\mathcal{C}} = \text{LLM}(\mathcal{C}) \in \mathbb{R}^d$, $f_{\hat{\mathcal{C}}} = \text{LLM}(\hat{\mathcal{C}}) \in \mathbb{R}^d$. Thus, the loss function for action difference descriptions can be computed as:

$$\mathcal{L}_{\text{comment}} = 1 - \text{cos\_sim}(f_{\mathcal{C}}, f_{\hat{\mathcal{C}}}). \tag{8}$$

The total loss function for model training is obtained by combining the three loss terms:

$$\mathcal{L} = \mathcal{L}_{\text{score}} + \alpha_1 \mathcal{L}_{\text{semantic}} + \alpha_2 \mathcal{L}_{\text{comment}} + \alpha_3 \mathcal{L}_{\text{KL}}, \tag{9}$$

where $\alpha_1$, $\alpha_2$ and $\alpha_3$ are coefficients controlling the weights of loss functions. During inference, we employ Eq 7 for difference description generation, without the need to have expert commentary as training.

## 4 EXPERIMENTS

### 4.1 EVALUATION METRICS & DATASETS

The proposed ADR task contains two sub-tasks: (1) score prediction, also known as action quality assessment (AQA), and (2) action difference description generation (ADDG). For AQA evaluation, we employ two standard metrics: Spearman's rank correlation ($\rho$) and relative $\ell_2$ distance R-$\ell_2$. The ADDG performance is assessed using established natural language generation metrics: BLEU Papineni et al. (2002), ROUGE Lin (2004) and METEOR Banerjee & Lavie (2005). Among these metrics, R-$\ell_2$ is unique in that lower values indicate better performance, while higher values of $\rho$, BLEU, METEOR and ROUGE signify superior performance. We test our model on three datasets: (1) our constructed ADR dataset for sports action assessment; (2) JIGSAWS dataset Gao et al. (2014) for surgery action assessment; (3) FitnessAQA dataset Parmar et al. (2022) for assessing fitness actions in gyms. The assessment of models on the FitnessAQA dataset is evaluated with F1 score following the practice of previous work Parmar et al. (2022). Detailed formulation of evaluation metrics and introduction of JIGSAWS and FitnessAQA datasets are contained in supplementary materials.

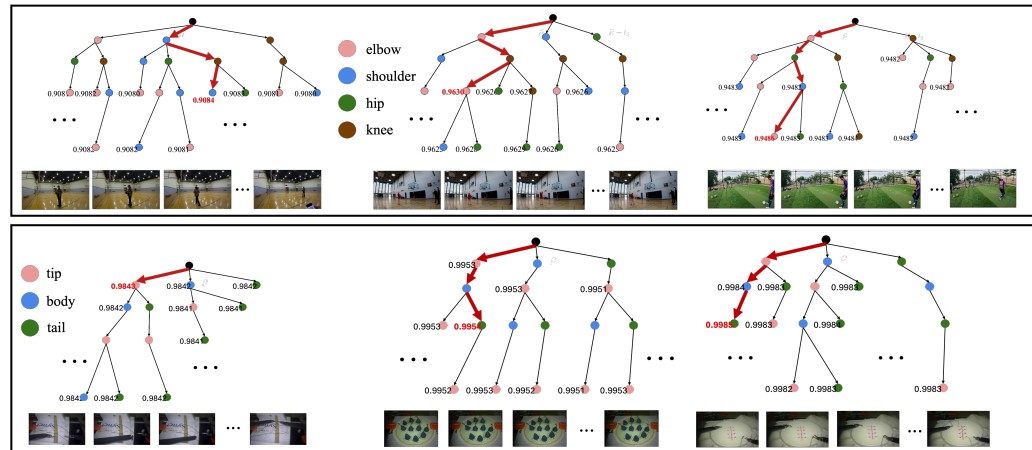

Figure 5: Visualization results of visual reasoning paths in keypoint-guided tree. Upper box: ADR dataset for sports actions; Lower box: JIGSAWS dataset for surgery actions. The selected reasoning paths are highlighted in red arrows. score values of top 10 paths are annotated in the figure.

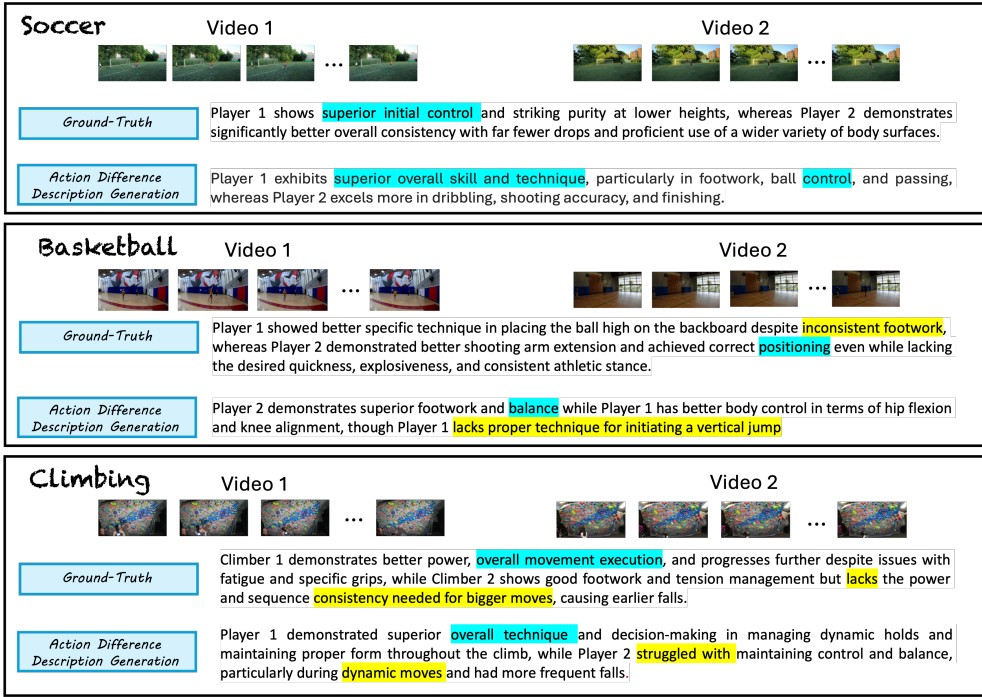

Figure 6: Qualitative results of generated action difference descriptions in our ADR dataset. Three sports categories (Soccer, Basketball, Climbing) with two videos forming a video pair. Highlighted texts indicates the similar semantic descriptions from our generated text with ground-truth.

## 4.2 QUANTITATIVE RESULTS

We compare our KEPT approach with existing approaches for Action Quality Assessment (AQA) and foundation models for action score prediction. Results in Table 2 and Table 3 demonstrate that our approach outperforms existing state-of-the-art approach for AQA in action score prediction. In addition, our approach outperforms both open source (LLaVA-Video Zhang et al. (2024), QWen 2.5 Yang et al. (2024)) and closed source (GPT-4o GPT (2024), Gemini-2.5 Gem (2025)) large vision-language models (LVLM). For open source LVLM, we finetune the model with LoRA Hu et al. (2022) on our ADR dataset. For action difference description generation, results in Table 4

| Models | ρ↑ | R − ℓ₂(×100)↓ |
|---|---|---|
| FineParser Xu et al. (2024b) | 0.41 | 29.36 |
| MAGR Zhou et al. (2024) | 0.73 | 10.14 |
| Qwen 2.5 Yang et al. (2024) | 0.46 | 44.26 |
| LLaVA-Video Zhang et al. (2024) | 0.21 | 16.93 |
| GPT-4o GPT (2024) | 0.54 | 11.87 |
| Gmini 2.5 Gem (2025) | 0.61 | 5.57 |
| Ours (Δ Expert Comment) | 0.64 | 5.17 |
| Our (Δ GNN) | 0.63 | 5.98 |
| Ours (Δ KL loss) | 0.77 | 4.34 |
| Ours (Δ Semantic Loss) | 0.59 | 6.62 |
| **Ours** | **0.80** | **3.55** |

Table 2: Action score prediction results in ADR dataset.

| Models | JIGSAWS | | FitnessAQA | | | |
|---|---|---|---|---|---|---|
| | ρ↑ | R − ℓ₂(×100)↓ | F1 (OK) | F1 (OE) | F1 (SF) | F1 (SI) |
| MAGR Zhou et al. (2024) | 0.48 | 10.601 | 0.564 | 0.357 | 0.796 | 0.157 |
| LLaVA-Video Zhang et al. (2024) | 0.41 | 40.614 | 0.392 | 0.533 | 0.795 | 0.257 |
| Gemini 2.5 Gem (2025) | 0.10 | 33.130 | 0.105 | 0.128 | 0.513 | 0.170 |
| GPT-4o GPT (2024) | 0.09 | 71.611 | 0 | 0.046 | 0 | 0 |
| Qwen-2.5-VL-7B Yang et al. (2024) | 0.03 | 67.475 | 0.404 | 0.082 | 0 | 0 |
| Ours (Δ KL Loss) | 0.50 | 10.654 | 0.598 | 0.447 | 0.477 | 0.219 |
| **Ours** | **0.67** | **7.469** | **0.670** | **0.502** | **0.812** | **0.400** |

Table 3: Action score prediction results in JIGSAWS and FitnessAQA datasets. OK: OHP_knee; OE: OHP_elbow; SF: Squat_forward; SI: Squat_inward.

demonstrates that our approach is able to generate better descriptions compared with competing approaches. We also compare action difference description generation results for each category in Table 9. Specifically, we evaluate four fitness actions: OHP knee, OHP elbow, Squar forward and Squat inward. Results in Table 3 indicate that our MCTS outperforms other approaches by a large margin on the FitnessAQA dataset accross all the action categories.

| Models | BLUE (n_gram) | | | | | ROUGE | | | | METEOR |
|---|---|---|---|---|---|---|---|---|---|---|
| | NG1 | NG2 | NG3 | NG4 | All | RO-1 | RO-2 | RO-L | RO-S | score |
| LLaVA-Video Zhang et al. (2024) | 0.168 | 0.016 | 0.002 | 0.002 | 0.009 | 0.15 | 0.011 | 0.106 | 0.106 | 0.144 |
| Gemini 2.5 Gem (2025) | 0.158 | 0.011 | 0.003 | 0.002 | 0.008 | 0.186 | 0.009 | 0.119 | 0.119 | 0.131 |
| GPT-4o GPT (2024) | 0.216 | 0.018 | 0.003 | 0.002 | 0.011 | 0.223 | 0.012 | 0.146 | 0.146 | 0.163 |
| Qwen 2.5 Yang et al. (2024) | 0.358 | **0.124** | **0.045** | **0.019** | 0.059 | 0.371 | 0.124 | 0.261 | 0.261 | 0.331 |
| Ours (Δ Semantic Loss) | 0.347 | 0.11 | 0.036 | 0.015 | 0.060 | 0.350 | 0.107 | 0.242 | 0.242 | 0.307 |
| Ours (Δ GNN) | 0.298 | 0.083 | 0.022 | 0.009 | 0.027 | 0.336 | 0.096 | 0.227 | 0.227 | 0.271 |
| Ours (Δ RKG) | 0.328 | 0.099 | 0.030 | 0.012 | 0.037 | 0.360 | 0.011 | 0.241 | 0.241 | 0.301 |
| Ours (Δ KL Loss) | 0.334 | 0.107 | 0.037 | 0.015 | 0.048 | 0.364 | 0.117 | 0.261 | 0.261 | 0.306 |
| **Ours** | **0.369** | 0.119 | 0.038 | 0.016 | **0.063** | **0.384** | **0.124** | **0.269** | **0.269** | **0.332** |

Table 4: Quantitative results for action difference description generation on ADR dataset. Δ RKG: residual keypoint graph. NG: n_gram; RO: ROUGE.

| Models | BLUE | | | | | ROUGE | | | | METEOR | ρ↑ | R − ℓ₂(×100)↓ |
|---|---|---|---|---|---|---|---|---|---|---|---|---|
| | NG1 | NG2 | NG3 | NG4 | All | RO-1 | RO-2 | RO-L | RO-S | score | | |
| Soccer | 0.399 | 0.116 | 0.032 | 0.013 | 0.058 | 0.391 | 0.125 | 0.268 | 0.268 | 0.334 | 0.93 | 1.51 |
| Basketball | 0.375 | 0.119 | 0.035 | 0.015 | 0.061 | 0.381 | 0.119 | 0.275 | 0.275 | 0.319 | 0.85 | 3.37 |
| Climbing | 0.367 | 0.120 | 0.040 | 0.017 | 0.064 | 0.373 | 0.120 | 0.257 | 0.257 | 0.321 | 0.74 | 5.21 |
| All | **0.369** | **0.119** | **0.038** | **0.016** | **0.063** | **0.384** | **0.124** | **0.269** | **0.269** | **0.332** | **0.80** | **3.55** |

Table 5: Quantitative results of our approach for action difference description generation and action score prediction on individual sports categories. NG: n_gram; RO: ROUGE.

## 4.3 Ablation Studies

Ablation studies in both Table 2 and Table 3 demonstrate the effectiveness and necessity of our designed modules and loss functions. Results in Table 2 indicate that removing semantic loss results in the most significant performance drop in terms of action score prediction. This suggests that language features from expert commentary is essential for guiding visual representations in decision making.

## 4.4 Qualitative Results

Figure 5 visualizes the learned reasoning paths from keypoint-guided tree in both sports and surgery scenarios. For those paths contain nodes belong to the same keypoint, we combine them into one single node. This visualization effectively demonstrates how our KEPT approach successfully models expert visual reasoning processes, capturing the hierarchical and sequential nature of domain-specific assessment protocols.. Figure 8 presents qualitative comparisons between generated action difference descriptions and ground-truth annotations. The highlighted text segments demonstrate that our generation framework effectively captures critical assessment information present in the ground-truth, validating the semantic fidelity of our approach in articulating visual differences that domain experts would identify as significant.

## 5 Conclusion

In this study, we addressed the challenging Action Difference Reasoning task, which demands both quantitative and qualitative reasoning capabilities. By formulating this task as a search problem, we develop the keypoint-guided tree search method to analyze performance differences through detailed kinematic assessment. Our experimental results demonstrate that this approach significantly outperforms existing methods in Action Difference Reasoning tasks. Future work will focus on extending our keypoint-guided tree search into a more comprehensive framework capable of handling increasingly complex reasoning challenges.

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

## A    APPENDIX

## B    RELATED WORK

**Tree Search.** Tree Search (MCTS) is a planning algorithm that strategically balances exploration and exploitation by integrating tree search with stochastic simulations Coulom (2006). The algorithm iteratively refines a search tree through four key stages. First, the selection phase traverses the tree from the root node, guided by a policy like Upper Confidence Bounds for Trees (UCT) Kocsis & Szepesvári (2006), to select a child node that either maximizes potential score or explores less visited states. Upon reaching a leaf node or an expandable internal node, the expansion phase adds one or more child nodes, representing previously unconsidered actions or plan extensions. Subsequently, the simulation (or rollout) phase estimates the value of the newly added node by performing random or policy-based sampling of actions until a terminal state or a predefined depth is reached. Finally, the backpropagation phase updates the value estimates of the nodes along the path from the newly simulated node back to the root, incorporating the simulation outcome. Through repeated iterations of these four phases, MCTS adaptively focuses its search on promising regions of the state space while maintaining exploration of less certain areas.

**Action Quality Assessment.** Action quality assessment has gained considerable attention recently, with diverse approaches aimed at enhancing accuracy and robustness. Wang et al. Wang et al. (2020) propose an uncertainty-aware score distribution learning framework, while Yang et al. Yang et al. (2021) introduce a group-aware contrastive regression model leveraging group dynamics for refined assessments. The TSA-Net architecture by Wang et al. Wang et al. (2021) uses tube self-attention to focus on relevant action segments, achieving improved performance. Additionally, FineDiving Wang et al. (2022a) provides a fine-grained, procedure-aware dataset for training models on nuanced action quality data. Further approaches include Wang et al.'s pairwise contrastive learning network Wang et al. (2022b), which emphasizes relationships between action pairs, and the FineParser model Wang et al. (2024a), which enables fine-grained spatio-temporal parsing for human-centric evaluations. The MAGR method Wang et al. (2024c) uses manifold-aligned graph regularization for continual assessment, adapting to evolving data distributions, while the GAIA framework Wang et al. (2024b) tackles the unique challenges of AI-generated video assessment. These studies collectively highlight the importance of robust frameworks and datasets in advancing action quality assessment. Recent work by Burgess et al. Burgess et al. (2025) introduced the video action differencing task, which compares how two individuals perform the same actions. However, their approach is limited to qualitative assessment only. In contrast, our work addresses both quantitative and qualitative assessment, enabling more comprehensive evaluation of performer differences.

## C    EVALUATION METRICS

(1) For score prediction on ADA and JIGSAW datasets, we utilize Spearman's rank correlation ($\rho$) and relative $\ell_2$ distance $R - \ell_2$ Pan et al. (2019); Parmar & Morris (2019) as evaluation metrics, while on the FitnessAQA dataset, we use F1-score as the evaluation metric:

- **Spearman's rank correlation ($\rho$)** focuses on testing sample ranking and is defined as :

$$\rho = \frac{\sum_{i=1}^{N}(y_i - \bar{y})(\hat{y}_i - \bar{\hat{y}})}{\sqrt{\sum_{i=1}^{N}(y_i - \bar{y})^2 \sum_{i=1}^{N}(\hat{y}_i - \bar{\hat{y}})^2}}, \tag{10}$$

  where $y$ and $\hat{y}$ indicate the ranking of two sequences. Higher $\rho$ suggests better model performance.

- **Relative $\ell_2$ distance** ($R - \ell_2$) measures the distance from each sample to the ground-truth:

$$R - \ell_2 = \frac{1}{N}\sum_{i=1}^{N}\left(\frac{|y_i - \hat{y}_i|}{y_{\max} - y_{\min}}\right), \tag{11}$$

  where $y_i$ and $\hat{y}_i$ denote the ground-truth and prediction scores for the $i$-th sample, respectively. Lower R-$\ell_2$ indicates better performance.

|  | $\rho \uparrow$ | $R - \ell_2 (\times 100) \downarrow$ |
|---|---|---|
| Ours (3 layers) | 0.61 | 5.11 |
| Ours (5 layers) | 0.68 | 4.31 |
| Ours (6 layers) | 0.58 | 6.26 |
| **Ours** (4 layers) | **0.67** | **7.469** |

Table 6: Ablation studies on the effect of different depth of tree search on score prediction performance.

- **F1-score** (F1) is the harmonic mean of precision and recall:

$$F1 = \frac{2TP}{2TP + FP + FN},\tag{12}$$

where

(2) For action difference description generation, we employ BLEU Papineni et al. (2002), ROUGE Lin (2004) and METEOR Banerjee & Lavie (2005) as evaluation metrics:

- **BLEU (Bilingual Evaluation Understudy)** measures how many words or phrases from a generated (candidate) translation overlap with a set of reference translations. It focuses on precision and considers n-grams to evaluate fluency and adequacy.

- **METEOR (Metric for Evaluation of Translation with Explicit Ordering)** is designed to address some of BLEU's limitations by considering recall alongside precision and focusing on word-level alignment. It also incorporates semantic similarity through synonyms and stemming.

- **ROUGE (Recall-Oriented Understudy for Gisting Evaluation)** is a family of metrics primarily used to evaluate automatic text summarization systems, though it also applies to machine translation. ROUGE compares automatically generated summaries or translations against one or more human-produced reference texts. All ROUGE metrics produce scores between 0 and 1, where higher values indicate greater similarity between the system output and human references.

## D   JIGSAWS AND FITNESSAQA DATASETS

**JIGSAWS Dataset.** The JIGSAWS dataset Gao et al. (2014) comprises recordings of three fundamental surgical tasks performed by surgeons on bench-top models, all commonly included in surgical training curricula. It contains three modalities: kinematic data, video recordings, and manual annotations. In this work, we focus on the video modality and associated surgical skill annotations.

**FitnessAQA Dataset.** FitnessAQA Parmar et al. (2022) includes three exercises: 1) BackSquat; 2) BarbellRow; and 3) Overhead (shoulder) Press. It also provides annotations from experts to evaluate the performance of athletes.

## E   MORE IMPLEMENTATION DETAILS

Feature dimension $d = 4096$. $\mathcal{F}_M, \mathcal{F}_V, \mathcal{F}_K, \mathcal{F}_H$ are all implemented with two-layer MLP. Multi-head attention consists of 8 heads. All experiments are conducted on two RTX-4090 GPUs.

## F   MORE QUANTITATIVE RESULTS

In Table 6 we present the ablation studies of implementing keypoint-guided tree with different depth. The results indicate that 4-layer keypoint-guided tree result in best score prediction performance. In Table 7 and Table 8, we present detailed results of score prediction performance on individual sports category (ADR dataset) and surgical activities (JIGSAWS dataset). We show detailed quantitative results of action difference description generation of individual sports category in Table 9.

| Models | Soccer | | Basketball | | Climbing | |
|---|---|---|---|---|---|---|
| | $\rho\uparrow$ | $R-\ell_2\ (\times100)\downarrow$ | $\rho\uparrow$ | $R-\ell_2\ (\times100)\downarrow$ | $\rho\uparrow$ | $R-\ell_2\ (\times100)\downarrow$ |
| MAGR Zhou et al. (2024) | 0.52 | 8.43 | 0.72 | 7.90 | 0.74 | 13.32 |
| FineParser Xu et al. (2024b) | 0.71 | 45.52 | 0.12 | 24.43 | 0.72 | 13.18 |
| LLaVA-Video Zhang et al. (2024) | 0.07 | 29.64 | 0.31 | 13.58 | 0.48 | 20.23 |
| GPT-4o GPT (2024) | 0.18 | 38.64 | 0.51 | 10.17 | 0.75 | 16.7 |
| Gemini 2.5 Gem (2025) | 0.64 | 14.21 | 0.63 | 7.01 | 0.70 | 4.34 |
| Qwen 2.5 Yang et al. (2024) | 0.70 | 69.49 | 0.48 | 11.37 | 0.43 | 59.67 |
| Ours ($\Delta$ Expert Commentary) | 0.57 | 8.72 | 0.77 | 4.68 | 0.56 | 6.37 |
| Ours ($\Delta$ GNN) | 0.48 | 7.26 | 0.75 | 5.99 | 0.54 | 7.14 |
| Ours ($\Delta$ KL loss) | 0.75 | 4.15 | 0.80 | 4.52 | 0.70 | 4.90 |
| Ours ($\Delta$ Semantic Loss) | 0.65 | 9.612 | 0.78 | 9.18 | 0.51 | 5.59 |
| **Ours** | **0.93** | **1.51** | **0.85** | **3.37** | **0.74** | **5.21** |

Table 7: Quantitative results of score prediction performance on the ADR dataset for the three sports categories (Soccer, Basketall, Climbing).

| Models | Knot | | Needle | | Saturing | |
|---|---|---|---|---|---|---|
| | $\rho\uparrow$ | $R-\ell_2\ (\times100)\downarrow$ | $\rho\uparrow$ | $R-\ell_2\ (\times100)\downarrow$ | $\rho\uparrow$ | $R-\ell_2\ (\times100)\downarrow$ |
| MAGR Zhou et al. (2024) | 0.45 | 8.84 | 0.56 | 13.37 | 0.42 | 9.60 |
| LLaVA-Video Zhang et al. (2024) | 0.45 | 31.74 | 0.17 | 48.69 | 0.61 | 41.41 |
| Gemini 2.5 Gem (2025) | 0.05 | 29.07 | 0.14 | 50.59 | 0.11 | 19.73 |
| GPT-4o GPT (2024) | 0.27 | 72.09 | 0.08 | 119.51 | 0.07 | 23.23 |
| Qwen 2.5 Yang et al. (2024) | 0.47 | 72.81 | 0.26 | 108.88 | 0.29 | 20.73 |
| Ours ($\Delta$ KL loss) | 0.54 | 10.61 | 0.61 | 12.07 | 0.36 | 9.28 |
| **Ours** | **0.72** | **6.81** | **0.81** | **7.44** | **0.49** | **8.17** |

Table 8: Quantitative results of score prediction performance on the JIGSAWS dataset for the three surgical activities (Knot, Needle, Saturing).

| Models | BLUE | | | | | ROUGE | | | | METEOR |
|---|---|---|---|---|---|---|---|---|---|---|
| | n_gram=1 | n_gram=2 | n_gram=3 | n_gram=4 | all | rouge-1 | rouge-2 | rouge-L | rouge-S | score |
| LLaVA-Video (Soccer) Zhang et al. (2024) | 0.155 | 0.015 | 0.002 | 0.002 | 0.007 | 0.143 | 0.006 | 0.093 | 0.093 | 0.126 |
| LLaVA-Video (Basketball) Zhang et al. (2024) | 0.163 | 0.017 | 0.002 | 0.002 | 0.009 | 0.136 | 0.005 | 0.099 | 0.099 | 0.145 |
| LLaVA-Video (Climbing) Zhang et al. (2024) | 0.169 | 0.016 | 0.002 | 0.002 | 0.009 | 0.154 | 0.012 | 0.011 | 0.011 | 0.143 |
| GPT-4o (Soccer) GPT (2024) | 0.238 | 0.021 | 0.006 | 0.006 | 0.013 | 0.235 | 0.016 | 0.160 | 0.160 | 0.171 |
| GPT-4o (Basketball) GPT (2024) | 0.220 | 0.017 | 0.003 | 0.003 | 0.011 | 0.225 | 0.014 | 0.147 | 0.147 | 0.158 |
| GPT-4o (Climbing) GPT (2024) | 0.212 | 0.018 | 0.003 | 0.002 | 0.011 | 0.221 | 0.012 | 0.145 | 0.145 | 0.165 |
| Gemini 2.5 (Soccer) Gem (2025) | 0.217 | 0.022 | 0.002 | 0.002 | 0.011 | 0.267 | 0.026 | 0.148 | 0.148 | 0.170 |
| Gemini 2.5 (Basketball) Gem (2025) | 0.163 | 0.010 | 0.003 | 0.002 | 0.008 | 0.180 | 0.010 | 0.117 | 0.117 | 0.130 |
| Gemini 2.5 (Climbing) Gem (2025) | 0.155 | 0.011 | 0.003 | 0.002 | 0.008 | 0.186 | 0.009 | 0.120 | 0.120 | 0.130 |
| QWen 2.5 (Soccer) Yang et al. (2024) | 0.359 | 0.154 | 0.064 | 0.027 | 0.054 | 0.394 | 0.139 | 0.291 | 0.291 | 0.339 |
| QWen 2.5 (Basketball) Yang et al. (2024) | 0.358 | 0.134 | 0.049 | 0.019 | 0.059 | 0.361 | 0.109 | 0.244 | 0.244 | 0.326 |
| QWen 2.5 (Climbing) Yang et al. (2024) | 0.357 | 0.102 | 0.035 | 0.018 | 0.054 | 0.371 | 0.127 | 0.268 | 0.268 | 0.333 |
| Ours ($\Delta$ Expert Commentary) (Soccer) | 0.364 | 0.096 | 0.024 | 0.008 | 0.045 | 0.371 | 0.102 | 0.261 | 0.261 | 0.311 |
| Ours ($\Delta$ Expert Commentary) (Basketball) | 0.366 | 0.107 | 0.034 | 0.014 | 0.058 | 0.377 | 0.114 | 0.266 | 0.266 | 0.306 |
| Ours ($\Delta$ Expert Commentary) (Climbing) | 0.342 | 0.109 | 0.039 | 0.018 | 0.063 | 0.368 | 0.116 | 0.253 | 0.253 | 0.321 |

Table 9: Quantitative results of different approaches for action difference description generation performance over individual sports categories on the ADR dataset.

# G   MORE QUALITATIVE RESULTS

We show more visualization results of the reasoning path in keypoint-guided tree in both the ADR dataset and JIGSAWS dataset in Figure 7. In Figure 8, we present more qualitative results of generated action difference descriptions in the ADR dataset.

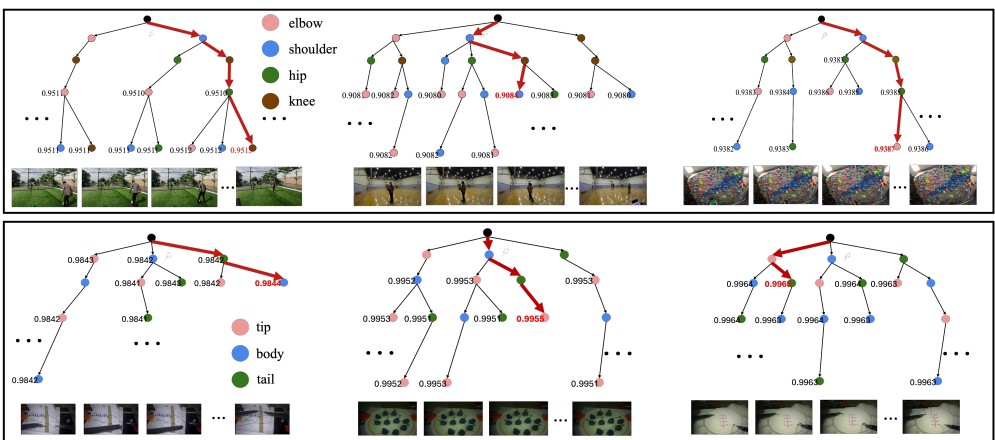

Figure 7: More qualitative visualization results of visual reasoning paths in keypoint-guided tree. Upper box: ADR dataset for sports actions; Lower box: JIGSAWS dataset for surgery actions. The selected reasoning paths are highlighted in red arrows. score values of top 10 paths are annotated in the figure.

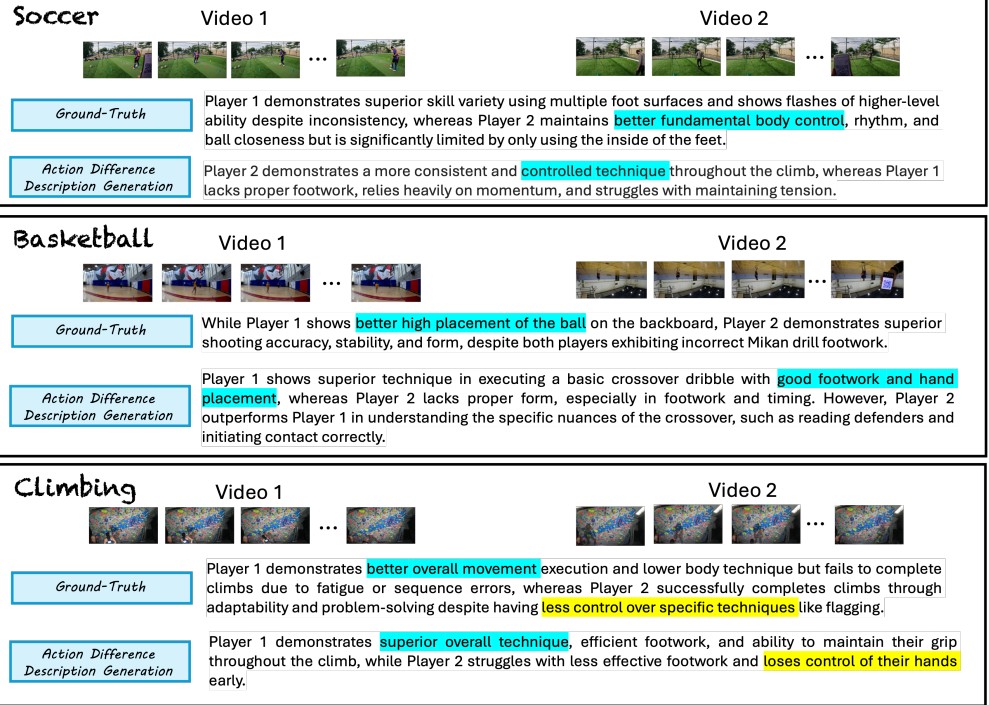

Figure 8: More qualitative results of generated action difference descriptions in our ADR dataset. Three sports categories (Soccer, Basketball, Climbing) with two videos forming a video pair. Highlighted texts indicates the similar semantic descriptions from our generated text with ground-truth.

