# OpenReview forum: "ActDR: Action Difference Reasoning via Keypoint Guided Tree Search"
_ICLR.cc/2026/Conference — ICLR 2026 Conference Withdrawn Submission_

### Official Review · Reviewer_1Eem · 2025-10-29

**Soundness:** 2
**Presentation:** 3
**Contribution:** 1
**Rating:** 2
**Confidence:** 4

**Summary:**

The paper introduces a new task, Action Difference Reasoning (ADR): given two videos depicting the same action, the model must output both a quantitative action-quality score and a qualitative, textual explanation of the differences. The authors construct a new ADR dataset (sourced from exocentric videos in Ego-Exo4D) and propose a keypoint-guided tree-search framework (KEPT). They claim state-of-the-art performance on both scoring and difference description, with additional generalization to standard AQA benchmarks (JIGSAWS, FitnessAQA).

**Strengths:**

- Idea of action differencing and reasoning is useful as shown by previous work.

- This work outperforms some baselines.

**Weaknesses:**

- Limited novelty. Pairwise comparison has been widely used in action assessment field, for example: [Refs 1,2,3 below]. This paper applies it to a different problem, which is marginal novelty. Furthermore, the architecture is very similar to [Ref 2 below]. Paper also conveniently does not discuss any of these work and comparisons.

- There have been many work in action quality assessment which offer qualitative assessment, for example, [Refs 4,5,6 below] to name a few.

- Paper does not discuss any related work. Moreover, all the references are mostly after year 2023.

- Tree search clarity. The method builds a limited-depth keypoint tree with joint angle/velocity/position features, uses cross-modal attention with the video to score each leaf path, and selects the highest-scoring path. Despite the terminology, this is closer to scoring and selecting from a fixed candidate set than to full MCTS; key components (e.g., UCT, rollouts, backpropagation of values), policy priors, and complexity analysis are missing. The paper should provide a bona fide search algorithm and details.

- Potential same-source bias. Ground-truth difference descriptions are generated by Gemini, while training/evaluation also rely on LLM semantic representations. The paper should introduce human-written references or task-based evaluation (e.g., “choose the better performer based on the description”) as primary metrics, or at least include sensitivity analyses across different LLM encoders/evaluators. Moreover, only 50 samples are checked and unreliably concluded that dataset is reliable.

- Fairness of closed-source VLM baselines. Clearly specify inference settings for proprietary LVLMs (temperature, context length, frame sampling, prompts, and cost constraints), and report zero-shot/few-shot/fine-tuning configurations for both open- and closed-source models to ensure fair longitudinal comparisons.

- Reproducibility and resources. Provide complete training details (optimizer, learning rate/schedule, batch size, sampling, input resolution), the keypoint extractor and 3D reconstruction pipeline, the effect of clip duration/frame count, and inference speed/compute cost.

- Other minor issues:

Line 165, Figure 1 should be Table 1.

Line 447, MCTS should be KEPT or KGT.


References:

[1] Jain, Hiteshi, Gaurav Harit, and Avinash Sharma. "Action quality assessment using siamese network-based deep metric learning." IEEE Transactions on Circuits and Systems for Video Technology 31.6 (2020): 2260-2273.

[2] Yu, Xumin, et al. "Group-aware contrastive regression for action quality assessment." Proceedings of the IEEE/CVF international conference on computer vision. 2021.

[3] Doughty, Hazel, Dima Damen, and Walterio Mayol-Cuevas. "Who's better? who's best? pairwise deep ranking for skill determination." Proceedings of the IEEE conference on computer vision and pattern recognition. 2018.

[4] Parmar, Paritosh, and Brendan Tran Morris. "What and how well you performed? a multitask learning approach to action quality assessment." Proceedings of the IEEE/CVF conference on computer vision and pattern recognition. 2019.

[5] Du, Zexing, et al. "Learning Semantics-Guided Representations for Scoring Figure Skating." TMM 2024.

[6] Zhang, Shiyi, et al. "Narrative action evaluation with prompt-guided multimodal interaction." Proceedings of the IEEE/CVF Conference on Computer Vision and Pattern Recognition. 2024.

**Questions:**

Please see weaknesses and may consider them as questions.

---

### Official Review · Reviewer_nSwa · 2025-10-31

**Soundness:** 2
**Presentation:** 2
**Contribution:** 2
**Rating:** 2
**Confidence:** 5

**Summary:**

This paper introduces a new computer vision task called Action Difference Reasoning (ADR). The goal of ADR is to not only assign quantitative performance scores to two similar actions performed by different individuals but also to generate a qualitative, natural language explanation of the key differences between them. The authors propose a framework named KEPT (Keypoint-guided Tree Search), which models the sequential visual reasoning process of a human expert as a search through a tree of possible keypoint-based observations. The framework consists of three main components: a keypoint-guided tree search to find the most salient reasoning path, an expert knowledge alignment module that uses textual expert commentary to guide the model, and an action difference reasoning module that generates both scores and textual descriptions. The authors constructed a new dataset, also called ADR, derived from the Ego-Exo4D dataset, which includes paired videos from sports like basketball, soccer, and rock climbing.

**Strengths:**

1.	The authors introduce a new task named Action Difference Reasoning (ADR) and it seems to help machines provide more reasonable feedback.
2.	The authors construct a new dataset named ADR which may contribute to the community.

**Weaknesses:**

1.	Lack of important comparative experiments. The author claims that the proposed method, KEPT, can be generalized to the AQA (Action Quality Assessment) settings and surpasses the state-of-the-art methods. However, the author did not conduct experiments on widely used AQA datasets in recent years (e.g., MTL-AQA[1], FineDiving[2]), and also lacks comparison with the latest methods since 2025. This reduces the credibility of the method.

[1] Parmar, P., & Morris, B. T. (2019). What and how well you performed? a multitask learning approach to action quality assessment. In Proceedings of the IEEE/CVF conference on computer vision and pattern recognition (pp. 304-313).

[2] Xu, J., Rao, Y., Yu, X., Chen, G., Zhou, J., & Lu, J. (2022). Finediving: A fine-grained dataset for procedure-aware action quality assessment. In Proceedings of the IEEE/CVF conference on computer vision and pattern recognition (pp. 2949-2958).

2.	Regarding the quality control of the dataset. Although the authors used Gemini to generate the action difference descriptions and conducted a sampling-based check, the reported consistency of only 85% with the ground-truth score differentials is concerning. This may lead to a low-quality dataset, potentially introducing significant noise that could affect the validity of the results.
3.	The presentation of the tables requires improvement. Some tables are scaled too small to be legible, particularly Tables 3 and 5. There is also a noticeable inconsistency in the scaling used for different tables, which negatively impacts the overall layout and readability of the manuscript.
4.	The figures in this paper are complex and cluttered, which makes them difficult to understand. For example, Figures 3 and 4 are particularly hard to interpret.
5.	Typo. Gmini in Table 2, Our in Table2.

**Questions:**

1.	Computational Cost. The paper utilizes a graph neural network (GNN) for Action Score Prediction, which may introduce more computational overhead and inference latency. Have the authors conducted any statistical analysis on this?
2.	Search mechanics. Do the authors use MCTS or beam search in Keypoint Guided Tree Search? Please detail selection if MCTS is indeed used.

**Details Of Ethics Concerns:**

None.

---

### Official Review · Reviewer_dEjZ · 2025-11-01

**Soundness:** 1
**Presentation:** 2
**Contribution:** 1
**Rating:** 2
**Confidence:** 4

**Summary:**

This paper presents the Action Difference Reasoning (ADR) task, which aims to compare two videos of similar actions to generate both performance scores and natural language descriptions of their differences. To address ADR, the authors propose KEPT, a framework that models expert-like reasoning by combining keypoint-based motion analysis with tree search to capture fine-grained kinematic cues. They introduce a new dataset, also called ADR, built upon the Ego-Exo4D dataset, with annotated video pairs from sports like basketball and soccer.

**Strengths:**

- The paper overall is easy to follow.

**Weaknesses:**

- My main concern is the value of the proposed tasks as well as the proposed solution compared to a simple cascade system. It is natural and well-motivated to want a model that can both produce accurate AQA score as well as provide a detailed explanation in natural language. However, I find the AQA performance of the proposed model is largely behind AQA models proposed in 2021. On JIGSAWS dataset, CoRe [r1] proposed in 2021 can achieve over 80%ρ while recent work like RICA² [r2] and MVLA [r3] can achieve ~90%ρ  which largely outperform the 67% result presented in the paper. So I think it is easy to outperform the proposed method by a cascade system with a 2021 model and any recent MLLM for producing explanations.

[r1] Group‑aware Contrastive Regression for Action Quality Assessment, ICCV 2021

[r2] RICA²: Rubric‑Informed, Calibrated Assessment of Actions, ECCV 2024

[r3] Vision‑Language Action Knowledge Learning for Semantic‑Aware AQA, ECCV 2024

**Questions:**

Please refer to my comments above. Overall, I think the authors fail to show the value the proposed task and model compared to a simple combination of two sub-tasks. I think the quality of the paper is clearly below the bar of ICLR.

---

### Official Review · Reviewer_y4XT · 2025-11-08

**Soundness:** 2
**Presentation:** 3
**Contribution:** 1
**Rating:** 2
**Confidence:** 5

**Summary:**

The paper addresses fine-grained differences in skilled activities like sports or surgery, introducing the Action Difference Reasoning (ADR) task and dataset based on Ego-Exo4D for quantitative scores and qualitative explanations of inter-performer differences. It proposes KEPT, a keypoint-guided tree search framework that captures kinematic cues, outperforming baselines on ADR and generalizing to traditional Action Quality Assessment (AQA) benchmarks like JIGSAWS and FitnessAQA.

**Strengths:**

- I agree with the authors' perspective that generating action difference scores alone is insufficient; it is also essential to produce action reasoning explanations.
- The paper is clearly presented with careful ablation studies.

**Weaknesses:**

- The proposed task is not novel and has been widely explored in recent years. For instance, works [1-2] have proposed methods to address this problem. The authors did not cite these papers and lack detailed comparisons with these methods.
- The proposed ADR dataset is essentially extracted from the existing Ego-Exo4D dataset with specific filtering and annotation. Thus, the claimed contribution of "introducing a new dataset" is not entirely convincing; it can be regarded more as an optimized version of an existing dataset.
- There is potentially significant noise in the data. The authors themselves noted that expert commentary is noisy. However, there is no human verification or cleaning beyond summarization with LLMs, which could introduce even more noise.
- Additionally, since there is no human verification of the ground truth annotations, one approach to validate the data could be to have typical humans (or ideally domain experts) perform the task and use their results as baselines. This would confirm that the data is clean and that humans can achieve strong performance on it. Such baselines are not provided, raising the possibility that the ground truth data is so noisy that the upper bound of the metrics is constrained by this noise.

[1]: Li Y M, Wang A L, Lin K Y, et al. TechCoach: Towards Technical-Point-Aware Descriptive Action Coaching[J]. arXiv preprint arXiv:2411.17130, 2024.

[2]: Ashutosh K, Nagarajan T, Pavlakos G, et al. ExpertAF: Expert actionable feedback from video[C]//Proceedings of the Computer Vision and Pattern Recognition Conference. 2025: 13582-13594.

**Questions:**

See Weaknesses

---

### Note · Authors · 2026-02-26

I have read and agree with the venue's withdrawal policy on behalf of myself and my co-authors.

---

### Meta-Review · Area_Chair_ms98 · 2026-01-10

**Summary:**

The reviewers reached a clear negative consensus on this submission with unanimous scores of 2. **Reviewer y4XT** notes that the ADR task lacks novelty compared to works like TechCoach or ExpertAF. These prior works are neither cited nor used as baselines. The reviewer also argues that the ADR dataset is simply a filtered subset of Ego-Exo4D with automatic annotations.

**Reviewer 1Eem** identifies major issues with the KEPT framework. The proposed tree search omits essential MCTS components like UCT, backpropagation, and policy priors. This reviewer also highlights a same-source bias because Gemini was used for both ground-truth generation and evaluation. Human verification was limited to only 50 samples, so the claims about data quality are not well-supported.

Empirical results are another major concern for **Reviewers dEjZ and nSwa**. The method achieves scores around 67%, while 2021 approaches like CoRe and MVLA reached 80–90%. The paper also omits experiments on standard AQA benchmarks such as MTL-AQA and FineDiving. **Reviewer dEjZ** suggested a baseline using an AQA model followed by an MLLM, but the authors did not provide this comparison.

The authors did not submit a rebuttal. As a result, none of these substantial concerns regarding novelty, dataset noise, or benchmarking were addressed. The consensus is that the work is not ready for acceptance.

**Reviewer Concerns:**

**Reviewer y4XT** focuses on the lack of task and dataset novelty. They argue that the ADR task has already been explored in TechCoach and ExpertAF. The authors did not cite these works or use them as baselines. Furthermore, the ADR dataset is viewed as a filtered version of Ego-Exo4D. Since LLMs generated the commentary without human verification, the supervision is likely noisy.

**Reviewer dEjZ** is concerned with the weak empirical performance. The proposed method performs far below established AQA approaches from 2021, which score between 80% and 90%. The reviewer also noted that a simple cascade of an AQA model and a modern MLLM would be a natural baseline. The authors did not implement or discuss this comparison.

**Reviewer nSwa** highlights the lack of evaluation on standard benchmarks like MTL-AQA and FineDiving. This makes it difficult to judge how the approach generalizes. They also point out that the ADR dataset has only 85% consistency with ground-truth score differentials. This level of noise affects the reliability of both training and evaluation.

**Reviewer 1Eem** questions the methodological soundness of the tree search. It lacks the core components required for a proper Monte Carlo Tree Search. They also raise concerns about the evaluation design and the heavy reliance on Gemini for ground truth. Missing details on training and computational costs further weaken the submission.

**Reviewer Scores:**

**Reviewer y4XT (Original: 2 → Predicted: 2):** The score is based on the lack of novelty and missing related work. No rebuttal was provided to address these gaps.

**Reviewer dEjZ (Original: 2 → Predicted: 2):** The reviewer noted clear empirical underperformance. Without new experiments or the requested cascade baseline, the score remains a 2.

**Reviewer nSwa (Original: 2 → Predicted: 2):** Concerns regarding dataset noise and missing standard benchmarks were not addressed.

**Reviewer 1Eem (Original: 2 → Predicted: 2):** The reviewer raised fundamental issues with the MCTS implementation and evaluation bias that remain unresolved.

---

### Decision · Program_Chairs · 2026-01-26

Reject